# (Optimal) Online Bipartite Matching with Degree Information

**Anders Aamand**
MIT
aamand@mit.edu

**Justin Y. Chen**
MIT
justc@mit.edu

**Piotr Indyk**
MIT
indyk@mit.edu

## Abstract

We propose a model for online graph problems where algorithms are given access to an oracle that predicts (e.g., based on modeling assumptions or on past data) the degrees of nodes in the graph. Within this model, we study the classic problem of online bipartite matching, and a natural greedy matching algorithm called MinPredictedDegree, which uses predictions of the degrees of offline nodes. For the bipartite version of a stochastic graph model due to Chung, Lu, and Vu where the expected values of the offline degrees are known and used as predictions, we show that MinPredictedDegree stochastically dominates *any* other online algorithm, i.e., it is optimal for graphs drawn from this model. Since the "symmetric" version of the model, where all online nodes are identical, is a special case of the well-studied "known i.i.d. model", it follows that the competitive ratio of MinPredictedDegree on such inputs is at least 0.7299. For the special case of graphs with power law degree distributions, we show that MinPredictedDegree frequently produces matchings almost as large as the true maximum matching on such graphs. We complement these results with an extensive empirical evaluation showing that MinPredictedDegree compares favorably to state-of-the-art online algorithms for online matching.

## 1 Introduction

Online algorithms are algorithms that process their inputs "on the fly", making irrevocable decisions based only on the data seen so far. Since they do not make any assumptions about the future, they are versatile and work even for adversarial inputs. Unfortunately, by focusing on the worst case, their performance in "typical" cases can be sub-optimal. As a result there has been a large body of research studying various relaxations of the worst-case model, where some extra information about the inputs, or the distribution they are selected from, is available [1].

Motivated by the developments in machine learning, over the last few years, many papers have studied online algorithms with predictions [47]. Such algorithms are equipped with a predictor that, when invoked, provides an (imperfect) prediction of some features of the future part of the input, which is then used by the algorithm to improve its performance. The specific information provided by such predictors is problem-dependent. For graph problems studied in this paper, predictions could include: the list of edges incident to a given vertex [33], the weight of an edge adjacent to a given node in an optimal solution [4], or vertex weights that guide a proportional allocation scheme [36].

In this paper we focus on online graph problems, and propose a model where an algorithm is equipped with a "degree predictor", i.e., an oracle that, given any vertex, predicts the degree of that vertex in the full graph (containing yet-unseen edges). This predictor has multiple appealing features. First it is simple, natural, and easy to interpret. Second, it is useful: vertex degree information is employed in many heuristic and approximation algorithms for graph optimization, for problems such as maximum independent set [24] or maximum matching [53]. Third (as demonstrated in Section 7)

36th Conference on Neural Information Processing Systems (NeurIPS 2022).

such predictors can be easily obtained. Finally, degree prediction is closely related to the problem of estimating the frequencies of elements in a data set[1], and frequency predictors have been already shown to improve the performance of algorithms for multiple data analysis problems [25, 28, 19, 17].

The specific graph problem studied in this paper is *online bipartite matching*, where we are given a bipartite graph $G = (U \cup V, E)$, and the goal is to find a maximum matching in $G$. In the online setting, the set $U$ is known beforehand, while the vertices in $V$ arrive online one by one. When a new vertex $v$ arrives, the edges in $G$ adjacent to $v$ are provided as well. Online maximum bipartite matching is a classic question studied in the online algorithms literature, with many applications [41]. It is known that a randomized online greedy algorithm, called Ranking, computes a matching of size at least $1 - 1/e \approx 0.632$ times the optimum [31], and that this bound is tight in the worst-case. A large body of work studied various relaxations of the problem, obtained by assuming that vertex arrivals are random [21] or that the graph itself is randomly generated from a given "known i.i.d. model" [20]. In this paper we extend the basic online model by assuming access to a predictor that, given any "offline" vertex $u \in U$, returns an estimate of its degree. (Note that the degree of any vertex in $V$ is known immediately upon its arrival.)

**Our results**  We study the following simple greedy algorithm for bipartite matching: upon the arrival of a vertex $v$, if the set of neighbors $N(v)$ of $v$ in $G$ contains any yet-unmatched vertex, the algorithm selects $u \in N(v)$ of minimum predicted degree in $G$ and adds the edge $(u, v)$ to the matching. This algorithm, which we call *MinPredictedDegree (MPD)*, is essentially identical[2] to the algorithm proposed in [10] which in turn was inspired by the offline matching algorithm called MinGreedy [53]. The intuition is that vertices with higher degree will have more chances to be matched in the future.

Our main contributions are as follows. First, following in a long line of work on the average-case analysis of matching algorithms initiated by [30], we analyze MPD under a natural random bipartite graph model we refer to as *CLV-B*, a bipartite version of the Chung-Lu-Vu random graph model [12]. A CLV-B random graph is parameterized by $n = |U|$, $m = |V|$, and two weight vectors $\mathbf{p} = \{p_i\}_{i=1}^n \in [0, 1]^n$ and $\mathbf{q} = \{q_i\}_{i=1}^m \in [0, 1]^m$. For any $u_i \in U$ and $v_j \in V$, the edge $\{u_i, v_j\}$ appears in the graph with probability $p_i q_j$ and these events are mutually independent. This model corresponds to the setting where consumers pick their edges with probabilities proportional to the vector $\mathbf{p}$ which describes the relative distribution over the producers.

Many natural families of random graphs can be described in the CLV-B model. Of particular interest is the case when $q = (1, \ldots, 1)$, corresponding to the consumers picking their edges i.i.d.; we will refer to this case as the *symmetric CLV-B model*. The symmetric version can be viewed as a special case of the well-studied known i.i.d. model of [20]. If we further let $\mathbf{p} = (p, \ldots, p)$, then the CLV-B graph is an Erdős-Rényi random bipartite graph with edge probability $p$.

**Theoretical Results**  For the CLV-B model and the MPD algorithm which uses the expected degrees as predictions, we make the following theoretical contributions:

- We show that MPD stochastically dominates *any* other online algorithm, i.e., it is optimal for graphs drawn from the CLV-B model (Section 5). Specifically, we show that for any degree distribution, any algorithm $A$ and any integer $t$, the probability that $A$ produces matching of size at least $t$ is upper bounded by the analogous probability for MPD. Since symmetric CLV-B is a special case of the known i.i.d. model, it follows that the competitive ratio of MPD for this model is at least equal to the best competitive ratio of any algorithm that works for the known i.i.d. model. By the result of [11], this ratio is at least 0.7299.

- We analyze MPD on symmetric CLV-B model with power law degree distribution (Section 6). Our theoretical predictions demonstrate that the competitive ratio achieved by our algorithm on such graphs is very high. In particular, for several different power law distributions, it exceeds 0.99.

- We also analyze MPD on Erdős-Rényi bipartite random graphs where all edges appear with the same probability (Appendix K). In particular, the competitive ratio of the algorithm on

---

[1]The degree of a node is simply the number of times the node appears in the union of all edges.

[2]The main differences are syntactic: the algorithm of [10] computes the degrees based on the given "type graph" (see Section 2), while in this paper we allow arbitrary predictors.

such graphs is at least $0.831$. Since in this case all expected degrees are equal, the prediction oracle is of no help. Thus, we conjecture that this is the worst distribution for MPD among all distributions in the CLV-B model class.

- Finally, we observe that the competitive ratio of MPD is $1/2$ for *worst case* graphs, and that this bound is tight. In addition, we show that the worst-case competitive ratio of *any* algorithm with access to the offline degrees is at most $1 - 1/e$, implying that degree predictions do not help in the worst-case though they prove to be useful in the random model as well as in practice. See Appendix E for details.

**Experiments**    We complement our theoretical studies with an extensive empirical evaluation of MPD for multiple random graph models and real graph benchmarks in Section 7. Our experiments show that, on most benchmarks, MPD has the best performance among about a dozen state-of-the-art online algorithms, even when compared to algorithms that use much more information about the input. These experimental results demonstrate that MPD performs well beyond the average-case instances we study theoretically.

**Prediction Error**    For our theoretical results on the CLV-B graphs, MPD is given only the expected (as opposed to the actual) degrees. Although this models the uncertainty in the input, it is natural to ask how MPD performs when even the expected degrees are mispredicted. To this end, in Appendix D, we suppose that the offline nodes are prioritized in an arbitrary order $\pi'$ which may be different from the order $\pi$ obtained by sorting the nodes according to their expected degrees. Letting $\Delta$ be the minimum number of offline nodes that needs to be deleted such that $\pi$ and $\pi'$ induce the same order on the remaining nodes, we prove that using a noisy degree predictor which induces $\pi'$ instead of $\pi$ can shrink the size of the matching produced by MPD by at most $\Delta$. We note that the number of mispredicted nodes is an upper bound on $\Delta$, but in general $\Delta$ could be much smaller.

Importantly, we also note that the empirical performance of MPD shows its resilience to prediction error. Our experiments on real graphs use predictors which are noisy and which degrade over time but still find large matchings. Furthermore, on synthetic Zipfian data, we experiment with artifically adding noise and show a gradual degradation of MPD's performance as error increases.

## 2   Preliminaries

**CLV-B model**    CLV-B is the bipartite version of the Chung-Lu-Vu model used in prior work [12, 42]. Given vectors $\mathbf{p} = \{p_i\}_{i=1}^n$ and $\mathbf{q} = \{q_j\}_{j=1}^m$, the edge $\{u_i, v_j\}$ appears in the graph independently with probability $p_i q_j$. From the vectors $\mathbf{p}$ and $\mathbf{q}$, we obtain the vector of offline expected degrees $\mathbf{d} = \{d_i\}_{i=1}^n = \{p_i \cdot \|\mathbf{q}\|_1\}_{i=1}^n$. For our theoretical results within this model, our algorithm MinPredictedDegree uses the degree predictor which returns the *expected* degree for each offline node: $\sigma(u_i) = d_i$ (see Appendix D for extension to noisy predictors). The particular case of symmetric CLV-B where $q = (1, \ldots, 1)$ corresponds to the case where consumers (online) pick their edges i.i.d. over producers (offline) and MPD has knowledge of the average preferences over producers.

**Known i.i.d. model**    In the known i.i.d. model of [20], algorithms are given access to a *type graph* $G = (U \cup V, E)$ and a distribution $\mathcal{P} : V \to [0, 1]$. The nodes in $V$ and their incident edges represent "types" of online nodes. An input instance $\hat{G} = (U \cup \hat{V}, \hat{E})$ is formed by picking $m$ online nodes i.i.d. from $V$ according to the probabilities described by $\mathcal{P}$. Note that the symmetric CLV-B model defined earlier is a special case of this model. In our experiments, the degree predictions are given by the expected degrees of the offline nodes.

## 3   Related Work

Online bipartite matching and its generalizations have been investigated extensively. The survey [41] and the recent paper [9] provide excellent overviews of this area. The state of the art competitive ratios are $1 - 1/e \approx 0.632$ in the worst case [31] and $\approx 0.7299$ for the known i.i.d. model [11]. See [9] for an extensive empirical study of the existing algorithms. Other algorithms examined in the experimental section include [20, 5, 40, 26, 18, 10].

---
**Algorithm 1** MinPredictedDegree
---
**Input:** Offline nodes $U$ and degree predictor $\sigma : U \to \mathbb{R}_{\geq 0}$
**Output:** Matching $M$
Initialize $M \leftarrow \emptyset$.
**while** online node $v \in V$ arrives **do**
    $N(v) \leftarrow$ unmatched neighbors of $v$
    **if** $|N(v)| > 0$ **then**
        $u^* \leftarrow \arg\min_{u \in N(v)} \sigma(u)$ (ties broken arbitrarily)
        $M \leftarrow M \cup \{(u^*, v)\}$
    **end if**
**end while**
---

More generally, there has been lots of interest in online algorithms with predictions over the last few years, for problems like caching [39, 50, 55, 29], ski-rental and its generalizations [49, 22, 2, 3], scheduling [45, 35] matching [33, 4, 36] and learning [14, 7]. Other areas impacted by learning-based algorithms include combinatorial optimization [13, 6, 15], similarity search [52, 56, 27, 54, 16], data structures [32, 44] and streaming/sampling algorithms [25, 28, 19]. See [47] for an excellent survey of this area.

## 4 Algorithm

**Online Bipartite Matching** The online bipartite matching problem is defined as follows. Given a bipartite graph $G = (U \cup V, E)$, we call $U$ the "offline" side and $V$ the "online" side of the bipartition. Let $n = |U|$ and $m = |V|$. The nodes in $U$ are known beforehand and the nodes in $V$ arrive one at a time, along with their incident edges. An online bipartite matching algorithm maintains a matching throughout the process, with the goal of maximizing the size of the matching. As each node $v \in V$ arrives, the algorithm can pick one of its neighboring edges to add to the matching.

**MinPredictedDegree** In addition to knowing the offline nodes $U$ beforehand, MinPredictedDegree (MPD) is given a degree predictor $\sigma : U \to \mathbb{R}_{\geq 0}$. In practice, this predictor could be inferred from additional knowledge about the graph or from past data. When a node $v \in V$ arrives, MPD (see Algorithm 1) uses this predictor to greedily select the minimum predicted degree neighbor $u^*$ of $v$ that is not already covered in the matching and then adds the edge $\{u^*, v\}$ to the matching. If no such valid neighbor exists, MPD does nothing with $v$. Intuitively, low degree offline nodes should be matched as early as possible as they only appear a few times while we will have many chances to match high degree offline nodes.

The MPD algorithm has similar structure to the worst-case optimal Ranking algorithm [31] which assigns a random cost to each offline node and at each step greedily matches with the lowest cost offline neighbor. Specifically, if the degree predictor is random, MPD and Ranking are equivalent. As we show in the later sections, if the predictor is "good enough", MPD often performs much better than Ranking, both in theory and in practice.

## 5 Optimality of MPD on CLV-B graphs

In this section we show that the size of the matching found by the the MPD algorithm stochastically dominates the size of the matching found by any other algorithm. We start by providing some preliminaries for the analysis.

**Preliminaries** For $p \in [0, 1]^n$ and $q \in [0, 1]^m$, let $I_{p,q}$ denote an instance of a CLV-B graph with $n = |U|$ offline nodes, $m = |V|$ online nodes, and weight vectors $p$ and $q$, such that the probability that an edge $(u_i, v_j)$ exists is equal to $p_i q_j$ for any $i \in [n], j \in [m]$. Assume with no loss of generality that $p$ is ordered, $p_1 \leq p_2 \leq \ldots \leq p_n$. Note that the expected degree of the offline node $u_i$ is $p_i \|q\|_1$, i.e., it is proportional to the weight $p_i$.

In the online setting, the nodes of $V$ arrive sequentially in the order $v_1, \ldots, v_m$ with the random neighborhood of $v_j \in V$ being revealed at the arrival of $v_j$. When $v_j$ arrives, an online bipartite

algorithm $A$ can match $v_j$ to any of its unmatched neighbors in $U$ but cannot change its decision later. For any online bipartite matching algorithm $A$, let $A(I_{p,q})$ denote the size of the matching attained by $A$ on the instance $I_{p,q}$. Let $A_0$ be the MinPredictedDegree algorithm which matches a node $v_j$ with neighborhood $S$ to an unmatched node $u_i \in S$ such that $p_i$ minimal, i.e. to an available node in $S$ with minimal expected degree (ties broken arbitrarily but consistently, e.g. by sorted order of the offline node id's). Let $p(A, S)$ be the resulting set of weights after algorithm $A$ (potentially) chooses a neighbor in $S$ to match with.

Consider two ordered weight vectors $p, p'$ both of length $n$. We say that $p'$ *dominates* $p$, equivalently $p \preceq p'$, if $p_i \leq p_i'$ for all $i \in [n]$. We are now ready to state our main result on the optimality of MPD.

**Theorem 5.1.** *Let $p \in [0, 1]^n$ and $q \in [0, 1]^m$. Let $A$ be any online algorithm and let $t \geq 0$. Then,*

$$\mathbb{P}(A(I_{p,q}) \geq t) \leq \mathbb{P}(A_0(I_{p,q}) \geq t).$$

To prove the theorem, we will need two technical Lemmas. Informally, Lemma 5.2 states that for any $S \neq \emptyset$, it is an advantage for $A_0$ if the neighborhood of the first arriving node is $S$ rather than the empty set. Lemma 5.3 (the proof of which is the main technical challenge) states that if $p \preceq p'$, then $A_0(I_{p',q})$ stochastically dominates $A_0(I_{p,q})$. Intuitively, Theorem 5.1 then follows from Lemma 5.3 by inducting on the number of online nodes $m$. For any algorithm $A$ and non-empty subset $S \subseteq [n]$, if $A$ matches $v_1$ to a node in the neighborhood $S$, then $p(A, S) \preceq p(A_0, S)$, and we can apply Lemma 5.3 together with the induction hypothesis with $m - 1$ online nodes. We need Lemma 5.2 to handle the issue that $A$ may not to match $v_1$ even in the case that $S$ is non-empty. The proofs of the two lemmas and of Theorem 5.1 are postponed to Appendices A, B and C.

**Lemma 5.2.** *Let $p \in [0, 1]^n$ and $q \in [0, 1]^m$ be weight vectors. Let $p^* \in [0, 1]^{n-1}$ be obtained from $p$ by removing its $i$'th entry for some $i \in [n]$. For any $t \geq 0$,*

$$\mathbb{P}(A_0(I_{p,q}) \geq t) \leq \mathbb{P}(A_0(I_{p^*,q}) \geq t - 1).$$

**Lemma 5.3.** *Let $p, p' \in [0, 1]^n$, be ordered weight vectors and $q \in [0, 1]^m$. Suppose that $p \preceq p'$. For any $t \geq 0$,*

$$\mathbb{P}(A_0(I_{p,q}) \geq t) \leq \mathbb{P}(A_0(I_{p',q}) \geq t).$$

While optimally only holds when the predicted degrees are the expected degrees (or at least induce the same ordering over the offline nodes), the performance of MPD cannot be much worse if the predictions are slightly off. Formally, for an arbitrary degree predictor $\sigma$, let $p[\sigma]$ be the array of CLV-B offline weights ordered by $\sigma$ and let $\text{LIS}(p[\sigma])$ be the size of the longest increasing subsequence in this array. We show (via a more general result) in Appendix D that MPD will match at most $n - \text{LIS}(p[\sigma])$ fewer nodes than when given the expected degrees as predictions.

# 6 Competitive ratio of MPD on symmetric CLV-B random graphs

Though we know that MPD is optimal within the CLV-B model, this result does not give explicit competitive ratios for MPD. In this section we analyze MPD under the symmetric CLV-B model, and derive a set of equations that give a lower bound on MPD's competitive ratio. To recap, the symmetric model is parameterized by $n = |U|$, $m = |V|$, and a vector $\mathbf{d} = \{d_i\}_{i=1}^n$ corresponding to the expected degrees of the offline nodes. Formally, for any $u_i \in U$ and $v_j \in V$, the edge $\{u_i, v_j\}$ appears in the graph with probability $d_i/m$.

As in the previous section, we analyze MPD when the degree predictions are given by the expected degrees $\mathbf{d}$. Our main results within this model are a set of equations that describe the size of the matching produced by MPD as well as the size of the maximum matching.

- Given a set of expected degrees $\mathbf{d}$, Equation 4 models the behavior of MPD on a symmetric CLV-B($\mathbf{d}$) graph. We extend these results to the asymptotic case in Appendix J, giving the expected matching size as $n, m \to \infty$ for a given distribution of expected degrees.

- Given a set of expected degrees $\mathbf{d}$,in Appendix I, we give an upper bound on the expected size of the maximum matching on a symmetric CLV-B($\mathbf{d}$) graph, and in Appendix J, we give the asymptotic equivalent. Empirically, we find this upper bound to be close to the maximum matching size when $\mathbf{d}$ follows a power law distribution.

- Using these equations, we show that in expectation MPD returns matchings almost as large as the maximum when the expected degrees of the offline nodes follow a power law distribution (see Table 1 and Figure 6). For both MPD and the maximum matching, we show that the matching sizes are concentrated about their expectations (Appendix L and M), implying that on these graphs, MPD achieves a large competitive ratio.

## 6.1 Competitive ratios on power law graphs

| Cutoff $\lambda$ | $\alpha = 0.5$ | $\alpha = 1$ | $\alpha = 1.5$ | $\alpha = 2$ |
|---|---|---|---|---|
| 10 | 0.967 | 0.948 | 0.934 | 0.928 |
| 100 | 0.998 | 0.986 | 0.958 | 0.937 |
| 1000 | 1.000 | 0.995 | 0.966 | 0.940 |
| 10000 | 1.000 | 0.997 | 0.969 | 0.940 |
| 100000 | 1.000 | 0.998 | 0.970 | 0.940 |

Table 1: Lower bound on the competitive ratio of MPD on symmetric CLV-B graphs with offline expected degrees following a power law with exponential cutoff distribution as $n, m \to \infty$. The fraction of offline nodes with expected degree $d$ is proportional to $d^{-\alpha} e^{-d/\lambda}$ for $d = \{1, 2, ...\}$.

In Table 1, we show the competitive ratio of MPD on symmetric CLV-B graphs with expected offline degrees following a power law with exponential cutoff distribution [9, 46] and with $n, m \to \infty$. For $d = \{1, 2, ...\}$, the fraction of offline nodes with expected degree $d$ is proportional to $d^{-\alpha} e^{-d/\lambda}$ for exponent $\alpha$ and cutoff $\lambda$. Note that in the asymptotic case, as the sizes of MPD's matching and the maximum matching are concentrated about their expectations (Theorems L.1, M.1), the ratio of expectations is equivalent to the competitive ratio (expectation of ratio). When the exponent is small or the cutoff is large, MPD achieves a better competitive ratio, with the ratio exceeding $0.99$ when both occur. When $\alpha = 2$, while MPD still achieves a competitive ratio of up to $0.94$, the competitive ratio is not as affected by a larger cutoff as with smaller exponents (the power law factor is already significantly limiting the fraction of offline nodes with large expected degree). The analysis we develop is general and can be used to evaluate MPD on symmetric CLV-B graphs with different parameters than those we have considered.

## 6.2 Differential equation analysis of MPD

Let $Y_d^t$ be the number of offline nodes with expected degree $d$ who are unmatched by MPD after seeing the $t$th online node. Within this random graph model, $\{Y_d^t\}_{t=0}^m$ form a Markov chain with the following expected evolution:

$$\mathbb{E}[Y_d^{t+1} - Y_d^t] = -\left(1 - (1 - d/m)^{Y_d^t}\right) \prod_{d' < d} (1 - d'/m)^{Y_{d'}^t}. \tag{1}$$

The first term corresponds to the probability that at least one unmatched offline node with expected degree $d$ is incident on the $(t + 1)$st online node while the second term corresponds to the probability that this online node has no neighboring unmatched offline nodes with smaller expected degree (which would be prioritized).

Let $k_d = -\log(1 - d/m)$. To simplify the analysis of MPD, it will be helpful to consider the random variables $Z_d^t = -k_d * Y_d^t$ where

$$\mathbb{E}[Z_d^{t+1} - Z_d^t] = k_d \left(1 - e^{Z_d^t}\right) \prod_{d' < d} e^{Z_{d'}^t}. \tag{2}$$

Following the work of Kurtz and many subsequent researchers [34, 57, 43, 38, 48], we show that the behavior of MPD as described by these Markov chains is well approximated by the trajectory of the following system of differential equations for all unique expected degrees $d$ in $\mathbf{d}$:

$$\frac{dz_d(t)}{dt} = k_d \left(1 - e^{z_d(t)}\right) \prod_{d' < d} e^{z_{d'}(t)}. \tag{3}$$

These functions $z_d(t)$ represent continuous-time approximations of the Markov chains with their derivatives corresponding to expected change from Equation 2. In Appendix G, we give the solution to these differential equations. Relying on past work [38], we give the following theorem (see Appendix H for proof).

**Theorem 6.1.** *Let $G$ be a symmetric CLV-B random graph with unique expected offline degrees $\{\delta_i\}_{i=1}^{\ell}$. Let $f_d = \lambda_d \cdot n$ be the number of offline nodes with expected degree $d$. Then, the expected (over the randomness in $G$) size of the matching formed by MPD approaches*

$$\sum_{i=1}^{\ell} f_{\delta_i} + z_{\delta_i}(m)/k \tag{4}$$

*as $n = m$ approach infinity, where $z_{\delta_i}(t)$ for $i \in \{1, \ldots, \ell\}$ form the solution to the system of differential equations in Equation 3.*

The solution to the system of differential equations gives us a closed form continuous-time approximation for expected performance of MPD in terms of **d**. In particular, in the asymptotic case, the equations give the exact expected performance and in the non-asymptotic case give an approximation on the number of unmatched offline nodes (and thus the matching size).

## 7 Experiments

In this section, we evaluate the empirical performance of MPD on real and synthetic data. For each dataset, we report the empirical competitive ratio of MPD and a variety of baselines. In each case, the empirical competitive ratio is the average, over 100 trials, of the ratios of the sizes of the matchings outputted by a given algorithm and the sizes of the maximum matching. In addition to the average ratio, we report one standard deviation of the ratio across the trials.

**Datasets**  We evaluate MPD on the following datasets.

- **Oregon:** 9 graphs[3] sampled over 3 months representing a communication network of internet routers from the Stanford SNAP Repository [37]. Each graph has $\sim 10k$ nodes on each side of the bipartition and $\sim 40k$ edges. For MPD, the offline degree predictor $\sigma : U \to \mathbb{R}$ is based on the first graph: if an offline node $u$ (i.e. a specific router) appeared in the first graph, $\sigma(u)$ is the degree of $u$ in that graph. If an offline node $u$ did not appear in the first graph, $\sigma(u) = 1$. For each trial, the order of arrival of the online nodes is randomized.

- **CAIDA:** 122 graphs[3] sampled approximately weekly over 4 years representing a communication network of internet routers from the Stanford SNAP Repository [37]. Each graph has $\sim 20k$ nodes on each side of the bipartition and $\sim 100k$ edges. The degree predictor is the same as for the Oregon dataset (for each year, the first graph of the year is used to form the predictor). As seen in Figure 8 (see Appendix N), the degree distribution of the graphs for both the Oregon and Caida datasets are long-tailed and the error of the first graph predictor increases over time as the underlying graph evolves. For each trial, the order of arrival of the online nodes is randomized.

- **Symmetric CLV-B random graph:** We consider symmetric CLV-B model where the expected offline degrees are distributed according to Zipf's Law, a popular power law distribution where $d_i = C \cdot i^{-\alpha}$ [46]. In our experiments, we set size $n = m = 1000$, set $C = m/2$, and vary the exponent $\alpha$.

- **Known i.i.d.:** Finally, we compare MPD to algorithms for the known i.i.d. model, copying the methodology of Borodin et al. [9] for synthetic power law graphs (Molloy Reed and Preferential Attachment) and real world graphs. In the Molloy Reed experiments, the type graph is sample from a family of random graphs with degrees distributed according to a power law with exponential cutoff. In the Preferential Attachment experiments, the type graph is formed by the preferential attachment model in which edges are added sequentially with edges between high degree nodes being more likely. The Real World graphs are comprised of a variety of graphs from the Network Repository [51]. See Appendix N for more results on Real World graphs.

---

[3]The graphs in the Oregon and CAIDA datasets are made bipartite following the *bipartite double cover* or *duplicating method* used in prior work [9]. Given a graph $G = (V, E)$, the bipartite double cover of $G$ is the graph $G' = (U' \cup V', E')$ where $U'$ and $V'$ are copies of $V$ and there is an edge $\{u'_i, v'_j\} \in E'$ if and only if $\{v_i, v_j\} \in E$.

**Baselines**   We compare our algorithm to a variety of baseline algorithms.

- **Ranking** In all experiments, we compare to the classic, worst-case optimal Ranking algorithm [31].

- **MinDegree** The MinDegree algorithm is a version of MPD with a perfect oracle, i.e. $\sigma(u)$ returns the true degree of $u$. In comparison with MPD, MinDegree shows the effect of prediction error on the performance of MPD.

- **Known i.i.d. baselines** For the experiments in the known i.i.d. case, we also compare to the baselines in the extensive empirical study of [9]–see their paper for detailed descriptions of all algorithms. The code is distributed under the GPL license. Notably, the algorithms Category-Advice and 3-Pass are *not* strictly online algorithms: they take multiple passes over the data, using some limited information from previous passes to make better decisions in the next pass. It should also be noted that BKPMinDegree is distinct from either the MPD or MinDegree algorithms we have described–it does not use the type graph but rather maintains and updates an estimate of the degree of the offline nodes throughout the runtime of the algorithm.

  Most known i.i.d. baselines are *not* greedy–they do not always match an online node even if it has unmatched neighbors. [9] evaluate greedy augmentations of these algorithms (denoted by Algorithm(g)) which match to an arbitrary unmatched neighbor in these cases and generally show them to outperform their non-greedy counterparts. We additionally evaluate MPD augmented versions of these algorithms (denoted by Algorithm(MPD)) which applies the MPD rule in these cases using the expected degrees as predictions.

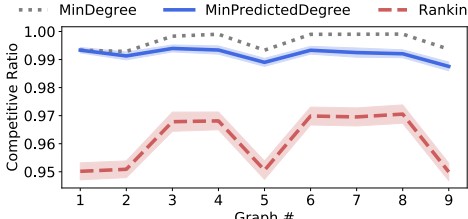

Figure 1: Comparison of empirical competitive ratios on the Oregon dataset. The first graph is used to form predictions.

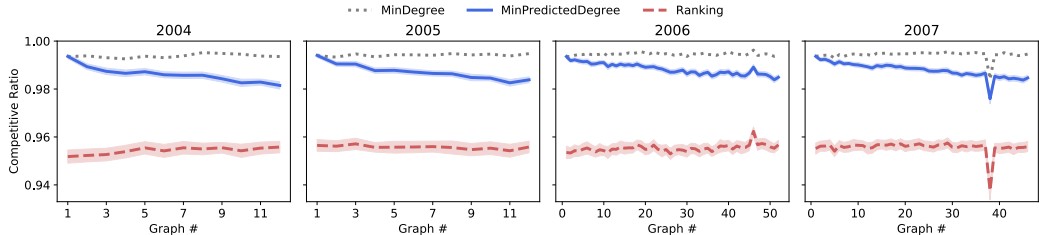

Figure 2: Comparison of empirical competitive ratios on the CAIDA dataset. For each subfigure, the first graph of the year is used to form predictions for the rest of the year.

**Results**   Across the various datasets, MPD performs well compared to the baseline algorithms. For the Oregon, CAIDA, and symmetric CLV-B random graph datasets, MPD significantly outperforms Ranking, and for Oregon and CAIDA, the performance of the algorithm mildly declines as the degree predictions degrade. For the known i.i.d. datasets, MPD often outperforms all *online* baselines, despite making only limited use of the known i.i.d. model. Additionally, augmenting the known i.i.d. algorithms with the (MPD) rule often improves their performance over both the base and the greedy (g) versions of the algorithms.

- **Oregon and CAIDA (Figures 1, 2):** On the Oregon dataset, MPD achieves a competitive ratio of $\sim 0.99$ across the graphs compared with competitive ratios ranging from $0.95$ to $0.97$ for

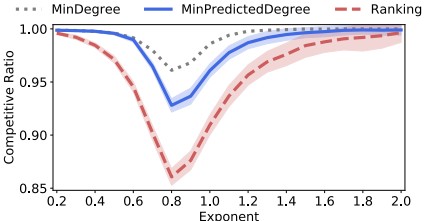
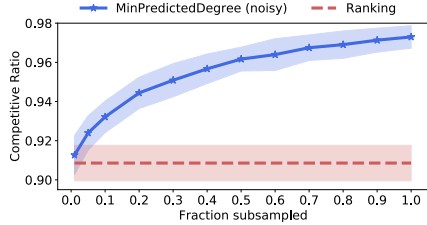

(a) Comparison across Zipf's Law exponents.

(b) Analysis of predictor noise (exponent $\alpha = 1$).

Figure 3: Comparison of empirical competitive ratios on symmetric CLV-B random graphs with offline expected degrees following Zipf's Law with exponent $\alpha$. In (a), we vary $\alpha$ and MPD uses the expected degree as its predictor. In (b), the degree predictor is the offline degree in a random subgraph using a (varying) fraction of the online nodes.

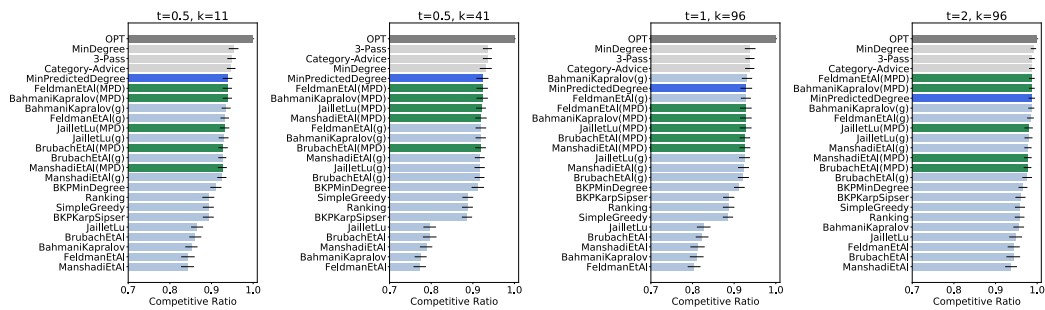

Figure 4: Comparison of empirical competitive ratios for Known i.i.d. Molloy-Reed graphs. Algorithms depicted in gray are *not* online algorithms (they use extra information or multiple passes). Algorithms in green are augmented with MPD.

Ranking. Compared with MinDegree, which uses knowledge of the true offline degrees, MPD's performance slowly degrades over time as the graphs become less similar to Graph #1 (see Figure 8 in Appendix N for quantitative details).

Similarly, on the CAIDA dataset, MPD does significantly better than Ranking, achieving competitive ratios almost always greater than $0.98$ compared to ratios around $0.95$, respectively. As the performance of the degree predictor degrades over time, the performance of MPD gradually declines (though it still significantly outperforms Ranking for both datasets).

- **Symmetric CLV-B random graph (Figure 3):** For symmetric CLV-B random graphs with offline expected degrees following Zipf's Law, MPD outperforms Ranking across a spectrum of exponents $\alpha$ ranging from $0.2$ to $2$. For exponents less than $0.5$ and greater than $1.5$, MPD achieves a competitive ratio close to $1$ (greater than $0.995$). All of the online algorithms have worse competitive ratios when the exponent is closer to one with MPD achieving a ratio of $\sim 0.93$ and Ranking achieving a ratio of $\sim 0.86$ when $\alpha = 0.8$. Though MPD does worst at $\alpha = 0.8$, it also achieves its greatest improvement over Ranking at this setting.

  In Figure 3b, we analyze the performance of MPD with a noisy degree predictor on Zipf's Law symmetric CLV-B random graphs with exponent 1. To introduce noise, the degree predictor $\sigma(u)$ is given by the number of neighbors $u$ has with a random subset of the online nodes $V$. As we decrease the fraction of $V$ we subsample, thus increasing the variance of the predictor, the performance of MPD steadily declines. Even when the degree predictor only uses 10% or even 1% (the leftmost point on the graph) of the online nodes, it still outperforms Ranking.

- **Known i.i.d. (Figures 4, 5):** Across all of the experiments in the known i.i.d. model, MPD is among the top online algorithms, and is often the best performing online algorithm (note the algorithms in gray are *not* strictly online algorithms). Most of the algorithms (e.g. BahamiKapralov and ManshadiEtAl) rely heavily on the type graph, including precomputing an optimal matching on the type graph. By contrast, MPD only uses first-order information: it only looks at degrees

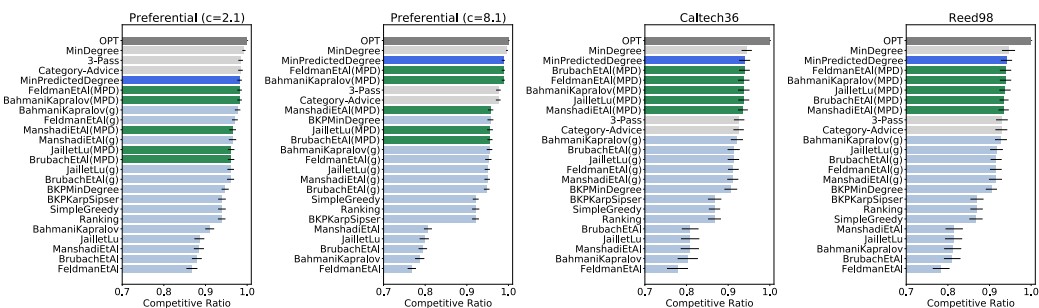

Figure 5: Comparison of empirical competitive ratios on Known i.i.d. Preferential Attachment graphs and Real World graphs. Algorithms depicted in gray are *not* online algorithms (they use extra information or multiple passes). Algorithms in green are augmented with MPD. See Appendix N for more Real World results.

and does not rely on any information about specific edges. Even so, in most cases, it outperforms all of the other online algorithms. Additionally, the (MPD) augmented versions of the known i.i.d. algorithms always beat the base algorithms and often beat the greedy (g) versions, indicating the potential of predicted degrees to be integrated with other algorithms. Note that while the standard deviations are quite wide (the known i.i.d. model is inherently stochastic), as the results are summarized over 100 trials, relatively small differences in the *average* performance of these algorithms are statistically significant as the standard error is small.

**Acknowledgements** This research was supported in part by the NSF TRIPODS program (awards CCF-1740751 and DMS-2022448), NSF award CCF-2006798, Simons Investigator Award, NSF Graduate Research Fellowship under Grant No. 1745302, MathWorks Engineering Fellowship, and DFF-International Postdoc Grant 0164-00022B from the Independent Research Fund Denmark.

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
