# OpenReview forum: "(Optimal) Online Bipartite Matching with Degree Information"
_NeurIPS.cc/2022/Conference — NeurIPS 2022 Accept_

### Official Review · Reviewer_gW3G · 2022-06-28

**Rating:** 4
**Confidence:** 3
**Soundness:** 3 good
**Presentation:** 3 good
**Contribution:** 2 fair

**Summary:**

This paper studies the problem of online bipartite matching. A new model called CLV-B is proposed under which the performance of the greedy algorithm is analyzed. For the general case, the paper proves that the greedy algorithm is optimal in terms of the success probability. For the special case, called symmetric CLV-B model, the paper analyzes the competitive ratio and the expected matching size, and further discusses the competitive ratio for power law and Erdos-Renyi graphs. Simulations are conducted to verify the practical performance of the proposed algorithm.



**Questions:**

To make the contribution clear, it is necessary for the paper to draw proper connections between the presented proofs and the existing proofs, or claim that the presented proofs are not inspired by any existing works.



**Limitations:**

- The paper does not discuss the limitations.
- One potential limitation is that some of the presented results can be easily acquired from existing papers.


**Strengths And Weaknesses:**

Strengths
- S1: The considered algorithm is definitely a natural choice for the CLV-B.
- S2: The paper presents a comprehensive study regarding the considered problem, and the presented results seem to be technically sound.
- S3: The experimental studies are also comprehensive, with source code available.

Weaknesses
- W1: The time and introduction suggest that the problem is solved using predicted variables, but the entire paper (module appendix D) focuses on the case where the exact expected degree is given. In this sense, the title, abstract, and introduction are somehow misleading, and could be improved to make it clear that this paper studies classic combinatorial optimization problems, without involving any machine learning or data-driven aspect. In this sense, the paper fits better with STOC/FOCS/SODA, even though NeurIPS has a wide scope.

- W2: My main concern is that the significance of the technical contribution is not very clear. The paper provides a lot of theoretical analysis (15 pages of proofs in the appendix), but it is not clear how easy or hard the presented results can be derived based on the existing works. For theory papers, the presented proofs are often inspired by some existing proofs, instead of being completely new. Such information is crucial for evaluating the significance of a new paper. For example, since the symmetric CLV-B model is a special case of the known iid model, does the analysis of the known iid model apply directly? What are the fundamental challenges caused by the CLV-B model in analyzing the algorithms?

- W3: The paper could be better organized such that all the theoretical results are clearly stated in the main paper. Many results in the appendix are only mentioned in the introduction but not discussed in the rest of the paper, and I am wondering whether such results are significant.

- W4: Some minors
	- In the case that q=1, it seems to be the case that the producers (not consumers) select their adjacent edges iid.
	- Instead of claiming that the competitive ratio exceeds 0.99 for power-law graphs, it would be better to simply state that the competitive ratio depends on the parameters of the power-law graph.
	- What does it mean by “the worst distribution for MPD”? It would be better to formally state the proposed conjecture.
	- P4, it would be better to give a formal definition of p(A, S).

---

> ### Author Response · Authors · 2022-08-01
> **Response to Reviewer gW3G (1/2)**
>
> Thanks for your review! We address several comments below.
>
> **W1:** We respectfully disagree with the characterization that the general degree prediction framework we introduce is misleading in terms of the content of the paper. In particular, our aim is to introduce the idea of general degree predictions for online bipartite matching (within the context of several online bipartite matching with predictions papers appearing in machine learning conferences). Our experiments (Section 7 and Appendix N) are extensive and in a variety of settings, including settings which clearly fall within the realm of data-driven algorithms with no assumptions about the distribution of graphs or the quality of predictions (see Figures 1 and 2) as well as in settings with added error (Figure 3b). Our theoretical results attempt to give some justification for the good empirical performance of MPD. To that end, we have to make some assumptions, and we focus on a constrained, but very natural, random setting of CLV-B graphs with expected degrees as prediction. *But this is by no means a limitation of the applicability of the algorithm in practice*–rather, we have to make some assumptions in order to effectively analyze the algorithm. As a final point, we emphasize that we analyze the case when the algorithm has access to the expected but not the true degrees (these quantities differ non-trivially).
>
> **W2:** In terms of distinguishing our technical contribution, we attribute any prior results used in our theorems while the rest of the math is our own. At a high level, our main results from Section 5 and 6 are non-trivial, and, while these use some known techniques (e.g., coupling random variables in Section 5), do not follow immediately from any prior work we are aware of. Specifically in relation to work on the known i.i.d. model, our analysis differs in that in every step, we make use of the specific structure of the CLV-B model (for instance, independence between edges). For completeness, we will go through all of our theoretical results and discuss their similarities to prior work in the following comment (see below).
>
> **W3:** Due to space, we had to make editorial decisions about what to include in the main text. We felt that having ample space to discuss Theorems 5.1 and 6.1 as well as the experiments was important but would be happy to consider any specific suggestions of what to move into the main text vs. the appendix.
>
> **W4:** Thanks for the writing suggestions, we will update the text accordingly. For the first comment, we indeed mean that the consumers (online nodes) pick their neighbors i.i.d.: each consumer $j$ forms an edge with producer $i$ w.p. $p_i$. On the other hand, the producers edges are not identically distributed (producers with high $p_i$ will have higher probabilities of forming edges). For the third comment, we mean to say that a setting of the parameters $\mathbf{p}$, $\mathbf{q}$ that minimizes the performance of MPD is one in which all $p_i$, $q_j$ take on the same value. We will make appropriate updates regarding the second and fourth points, thanks.

---

> > ### Author Response · Authors · 2022-08-01
> > **Response to Reviewer gW3G (2/2)**
> >
> > Here is a catalogue of all of our theoretical results and their relation to prior work in response to W2.
> > - Theorem 5.1, Appendices A, B, C (Optimality of MPD): To our knowledge, this proof idea is novel, and we did not rely on any prior work for this result. At a very high level, the general structure of the argument follows a “coupling” argument which is a standard technique to prove stochastic dominance.
> > - Theorem 6.1, Appendices G, H, J (Competitive ratio analysis via differential equations): As discussed in both the exposition and proof itself, the core tool used in lower bounding the size of the matching returned by MPD is a result on relating continuous differential equation approximations to the asymptotic behavior of random discrete processes (see, for instance, the cited works of Kurtz or Wormald). However, finding the right system of equations to describe our problem on which this technique can be applied as well as finding a solution to those equations was, to our knowledge, original.
> > - Appendix D (Analysis of MPD with noisy predictions): This analysis is original, and we don’t know of any closely related prior work.
> > - Appendix E (Worst-case bound): The construction of the worst-case example for MPD with true degrees as predictions is a construction which uses the hard example of Karp, Vazirani, and Vazirani. However, this specific construction applied to our problem with degree information is, to our knowledge, novel.
> > - Appendices I, J (Upper bound on maximum matching size): We came up with our approach to upper bounding the size of the maximum matching independently of prior work and do not know of any other work which uses this specific idea though it is certainly true that the general technique of finding sets which violate Hall’s theorem is a common idea in matching theory.
> > - Appendix K (Analysis in the special case of Erdos-Renyi bipartite graphs): We came up with this analysis independently of prior work by, in part, applying our general analysis from Section 6 to this special case. However, matchings in Erdos-Renyi graphs have been intensively studied, and after extensive search for related work, we have found similar results on greedy matchings in Erdos-Renyi bipartite graphs using a refined upper bound on the maximum matching specific to the special setting where n=m from Mastin and Jaillet 2013: https://www.mit.edu/~jaillet/general/greedy_bip.pdf. This paper gives an improved result on the competitive ratio in this conjectured hard case of 0.837 over 0.831. We will cite this paper as an improvement over our results in the balanced case (n=m) but note that our results generalize to the unbalanced case. In fact, analysis of the unbalanced case was stated as possible future work in that paper.
> > - Appendices L and M (Concentration of the quantities from section 6): For these results, the underlying idea is the standard technique of relating the properties of interest to a martingale and applying Azuma’s inequality. The specific application to our problem is our own.

---

> > > ### Author Response · Authors · 2022-08-07
> > > **Follow-up to reviewer gW3G**
> > >
> > > Dear Reviewer gW3G,
> > >
> > > Did we address all your concerns satisfactorily? If your concerns have not been resolved, could you please let us know which were not sufficiently addressed so that we have a chance to respond? Thanks!

---

> > > > ### Comment · Reviewer_gW3G · 2022-08-07
> > > > **To authors**
> > > >
> > > > Thanks for your response.
> > > >
> > > > First, I would like to keep my opinion that this paper does not fit very well with Neurips. Except for the experiments, the problem studied in this paper is purely combinatorial optimization.
> > > >
> > > > Second, I have the following two *optional* questions:
> > > >
> > > > 1 The proposed algorithm is the same as the Ranking algorithm of Karp, Vazirani and Vazirani proposed in 1990. Could you compare your optimality results with the one analyzed in the paper of Karp el al.?
> > > >
> > > > 2 The online bipartite matching problem has been extensively studied following the seminal work of Karp, and the algorithms and settings in this paper do not differ much from the existing works. I understand that the analysis in this paper may be non-trivial, but I am a little surprised that the authors claim that many of the results are novel. In that sense, the paper seems to be a very strong contribution to theoretical computer science, as online bipartite matching is a fundamental problem. In the authors' opinion, if the paper were submitted to STOC/FOCS/SODA, what is the chance that this paper can be accepted?
> > > >
> > > > Thanks,

---

> > > > > ### Author Response · Authors · 2022-08-08
> > > > > **Reply to reviewer gW3G**
> > > > >
> > > > > ### 1) Ranking algorithm
> > > > >
> > > > > It is not true that our algorithm is the same as the Ranking algorithm. They are related in that they both choose which node to match greedily based on an ordering of the offline nodes. However, in Ranking the ordering is random, and in our algorithm, it is given by a set of predictions. We remark that such alterations can affect the performance of the algorithms significantly. For a related example, if instead of using Ranking, one matches each online node independently to a random offline node in its neighborhood, the competitive ratio drops from 1-1/e to 1/2. Therefore, the analysis of our algorithm would not apply at all to the Ranking algorithm as we are leveraging the fact that the predictions are related to degrees. Likewise, the analysis of Ranking does not seem to apply in our setting. This is validated by our experiments where our algorithm performs much better than Ranking.
> > > > >
> > > > > ### 2) Choice of venue
> > > > >
> > > > > In terms of fit with NeurIPS, many papers on algorithms with predictions (see our Related Work section for some examples) have appeared at NeurIPS and other machine learning conferences where the core technical analysis is concerned with combinatorial optimization. We think that this paper would be of interest to that community.
> > > > >
> > > > > For this reason as well as because of our extensive experiments–which we think are a key part of this paper, we think that a machine learning conference is the most appropriate venue for this work.
> > > > >
> > > > > ### "I am a little surprised that the authors claim that many of the results are novel."
> > > > >
> > > > > If the reviewer is aware of any papers with results that overlap with ours, we would appreciate the references.

---

### Official Review · Reviewer_FyN6 · 2022-07-07

**Rating:** 7
**Confidence:** 3
**Soundness:** 4 excellent
**Presentation:** 3 good
**Contribution:** 4 excellent

**Summary:**

This paper studies the online bipartite matching problem. The graph edges are not known beforehand but arrive in a certain order. Given Graph G(U\cup V, E), U is known while the nodes in V arrive online. Once a node in V arrives, its incident edges are also known. The paper aims to find the maximum matching in CLV-B model graph. The CLV-B model assigns a probability value to each node, and an edge linking from one node in U and one node in V exists with probability that is the product of the probabilities of the two nodes.

The main contributions of the paper are new theories and proofs that show a classic MinPredictedDegree (MPD) based method can achieve optimum in the CLV-B model. It also further analyzes the lower-bounds of the competitive ratios in several special cases.

Overall, the contributions look solid to me and the paper is well written.

**Questions:**

Q1. In fig 2, why this is a fluctuation in 2007?

Q2. I cannot get the intuition behind Theorem 6.1 (Eq. 4). Please explain the results with more intuition.

Q3. Since the a CLB-V graph is a special case of a general graph, and there are results for the online bipartite matching for the general graphs already. I wonder how difficult are the proofs in addition to the known results.

**Limitations:**

The results of this paper is very interesting. However, the link of the studied problem to the community of Machine Learning is not that clear. How is the online bipartite matching applied in some ML based applications? If the link not that strong, I am not sure NeurIPS is the best fit for this paper.

**Strengths And Weaknesses:**

S1. This paper shows a theoretical contribution on a classic problem -- online bipartite matching.

S2. This paper is in general well written.

S3. It also analyzes the bias when the degree predictor has some errors.

S4. It analyzes the lower-bounds of the competitor ratios in some special cases.

S5. It shows empirical competitive ratios.

W1. This paper assumes that there is an oracle that can predict the degree for any node in U. In CLV-B model, the expected degree is used.

W2. The link of this problem to the community of Machine Learning should be made clearer. Is a pure theory conference better for this paper?

---

> ### Author Response · Authors · 2022-08-01
> **Response to Reviewer FyN6**
>
> Thanks very much for your review! We address several comments and questions below.
>
> **W2:**  As described in the Related Work section, multiple prior papers on online matching with predictions have appeared in machine learning conferences. Furthermore, our paper includes extensive experiments (Section 7 and Appendix N) for a variety of prediction methods. Thus, we believe that a machine learning conference is the most appropriate venue for this work.
>
> **Q1:** While we are not sure of the exact underlying cause, the graph from that specific day is much smaller (around 1/3 the size) than the other graphs and is an outlier in the sequence.
>
> **Q2:** At a high level, the differential equation analysis works as follows. First, we consider the number of offline nodes of a given degree $d$ that are matched at the $t$th timestep (upon seeing the $t$th online node). This is a discrete random process that we can model as a Markov chain, but it is difficult to analyze its behavior. Borrowing from prior work, we show that this discrete process can be asymptotically approximated by a continuous version where the time variable is continuous. In that case, these differences between timesteps can be viewed as differential equations. By getting a closed form solution to the system of differential equations, we can then understand the expected behavior of the algorithm after all $n$ timesteps. Does this help explain these results? Happy to add more details if not.
>
> **Q3:** In general, our results (in particular, Theorems 5.1 and 6.1) are tailored to the specific properties of the CLV-B graphs such as independence between edges and do not follow from prior work in other settings. We are not aware of optimality results similar to Theorem 5.1 on other variants of the matching problem. The differential equation technique is a known technique, but finding the right system of equations to describe our problem on which this technique can be applied as well as finding a solution to those equations is original to our knowledge. See the response to Reviewer gW3G below for a complete description of how our techniques relate to prior work.

---

> > ### Comment · Reviewer_FyN6 · 2022-08-07
> > **To authors**
> >
> > Thanks for the response. My concerns are well addressed. The authors can consider adding the explanations in Q2 into the paper.

---

### Official Review · Reviewer_Skmv · 2022-07-10

**Rating:** 7
**Confidence:** 4
**Soundness:** 4 excellent
**Presentation:** 4 excellent
**Contribution:** 4 excellent

**Summary:**

This work considers the online bipartite matching problem with an oracle that predicts the degrees of nodes in the graph. They proposed the MinPredictedDegree(MPD) algorithm, which matches each online node to the offline node with the minimum predicted degree.

On a natural random graph model (CLV-B), the expected degrees of the offline nodes are known and used as predictions. They showed that this MPD algorithm stochastically dominates any other online algorithms.

They analyze this algorithm on a natural random graph model (CLV-B) with power-law degree distribution. Using the expected degrees as the predictor in the algorithm, they show that the competitive ratio of this algorithm to the maximum offline matching is more than 0.94 for several model parameters.

The experimental results on multiple random graph models and real-world graphs show that the performance of MinPredictedDegree exceeds state-of-the-art online algorithms.

**Questions:**

1. In the introduction and appendix D, you showed that the MDP is robust to the prediction error. I think your proof is the robustness of the MDP on the CLV-B graphs, right? I am confused about this proof where you used that the graph is a CLV-B random graph. Is a similar analysis of the robustness also hold for other graphs?

**Strengths And Weaknesses:**

Strengths:
Learning augmented algorithms is a popular and important topic. Online bipartite matching with degree predictors is a new and interesting setting.

This work analyzed the MinPredictedDegree algorithm for this problem. They showed the optimality of the MPD algorithm on the CLV-B random graphs. The theoretical analysis of this algorithm on the random graph (CLV-B) model is also very interesting. On CLV-B random graphs with power-law degree distribution and many real-world datasets, they show that the MPD algorithm achieves a substantial improvement.

This paper is well-written and easy to follow.

I reviewed this paper before, they improved the results by adding the optimality analysis.

---

> ### Author Response · Authors · 2022-08-01
> **Response to Reviewer Skmv**
>
> Thanks very much for your (second) review! In response to your question on the robustness result, the one place we use CLV-B graphs is in using Lemma 5.2 which states that for MPD with expected degrees, removing an offline node never decreases the matching size by more than one. As you suggest, this statement should hold for MPD run on an arbitrary graph with an arbitrary degree predictor, so the result should generalize to any fixed graph (and therefore any distribution over fixed graphs). In particular, for any graph G and two degree predictors $\sigma$ and $\sigma’$, the size of the matchings returned by MPD run using the two degree predictors differ by at most the minimum number of nodes which must be deleted for the resulting permutations induced by $\sigma$ and $\sigma’$ to be the same. Thanks for the suggestion, we will update the statement and proof.

---

> > ### Comment · Reviewer_Skmv · 2022-08-08
> > **To authors**
> >
> > Thanks for the response and clarification. My question is well addressed.

---

### Official Review · Reviewer_o3xo · 2022-07-11

**Rating:** 6
**Confidence:** 4
**Soundness:** 3 good
**Presentation:** 4 excellent
**Contribution:** 3 good

**Summary:**

This paper considers the online bipartite unweighted matching problem with predicted degrees of offline vertices. The author proposes a simple algorithm MPD which matches the vertices with low degrees first to exploit the predicted degree. This algorithm has a sub-optimal 1/2 competitive ratio in the worst case. But it performs well on some real data set and CLV random graphs, as the experiments in this paper show. The author proves that this algorithm is optimal on CLV-B random graphs when the prediction is perfect. The author also estimates the lower bound of the competitive ratio of MPD on symmetry CLV-B random graphs through solutions of some derivative equations.

**Questions:**

1.	In the paragraph “On prediction error”, the author uses the quantity “the minimum number of offline nodes that needs to be deleted such that π and π_0 induce the same order on the remaining nodes” to measure the prediction error. But in Appendix D, the author uses the quantity “Largest Increasing Subsequence”. It seems that these two are the same thing, but it’s better to use the same expression to avoid confusing.
2.	The optimality result assume that the prediction is perfect in fact. But a “prediction” itself means it is not perfect. If all the theoretic results (except Appendix. D) assume that the prediction is perfect, then it might be better to consider the setting such as “xxx with known expected vertex degrees” but not “xxx with predicted degrees”.


**Limitations:**

Yes. I don’t think there are potential negative societal impact of their work.

**Strengths And Weaknesses:**

Strengths:

1.	Online bipartite matching problems are very important and have many applications in the real world. The prediction model in the paper is simple and well-motivated. It’s reasonable that we can obtain predicted degrees in real-world applications.
2.	The experimental results are well done and detailed. The author compares MPD and Ranking on real-world data sets, symmetry CLV-B graph and known i.i.d setting. The MPD outperforms Ranking well. And it seems that even when the degrees are estimated very poor, MPD still outperforms Ranking on CLV-B random graphs.
3.	Though MPD is sub-optimal in the worst case, the author shows that it is optimal on CLV-B random graphs. This is a nice theoretical result.

Weakness:

1.	The theoretical results are not general enough. They only cover the CLV random graphs where the expected degrees obey zipf’s law.
2.	Main result on the optimality of MPD assumes that the predictions are perfect. If “prediction” is the main motivation of this paper, the error of the prediction is an important factor to study. However, the paper does not discuss the performance under imperfect prediction thoroughly. This makes the contribution of the paper limited.

---

> ### Author Response · Authors · 2022-08-01
> **Response to Reviewer o3xo**
>
> Thanks very much for your review! We address several comments and questions below.
>
> **W1. “They only cover CLV random graphs where the expected degrees follow zipf’s law.”**
> To clarify, while our theoretical results are concerned with CLV-B random graphs, we study them in their full generality, not just when the distribution of degrees follows Zipf’s law. For instance, Theorem 5.1 shows optimality of MPD on any CLV-B random graph given access to the expected degrees, regardless of the parameters of the random graph. The equations in Theorem 6.1 apply to MPD run on any symmetric CLV-B random graph with any distribution of parameters. We plug in the specific Zipf’s laws parameters into these equations to demonstrate how this general analysis can be applied to show how MPD performs on this popular and useful specific distribution.
>
> **W2. Prediction error**
> Our experiments (Section 7 and Appendix N) demonstrate that MPD performs very well even when given imperfect predictions from past data which have no guarantees (see Figures 1 and 2), when given much weaker predictions than other algorithms (in the known i.i.d. setting), and in the CLV-B setting with added prediction error (Figure 3b). In our theoretical analysis, we assume access to the expected not actual degrees of the offline nodes and emphasize that these quantities can differ significantly. Finally, we do theoretically address the issue of errors in predicting the expected degrees in Appendix D though we agree that further exploration of this issue would be interesting future work.
>
> **Q1**
> Thanks for pointing out this potential point of confusion. The two quantities are complements of each other. We will decide on a single expression (perhaps while mentioning the alternative in passing) in the final text.
>
> **Q2**
> Regarding prediction error, see the response to W2 above. In terms of the overall setting of the problem, our aim is to introduce the idea of degree predictions for online bipartite matching without regard to a specific type of prediction (within the context of several recent online bipartite matching with predictions papers). Our experimental results are carried out in this general setting, and our theoretical results attempt to give some justification for the good empirical performance of MPD. To that end, we have to make some assumptions, and we focus on a constrained, but very natural, random setting with expected degrees as prediction. *But this is by no means a limitation of the applicability of the algorithm in practice*–rather, we have to make some assumptions in order to effectively analyze the algorithm.

---

> > ### Author Response · Authors · 2022-08-08
> > **Follow-up to Reviewer o3xo**
> >
> > Dear Reviewer o3xo,
> >
> > Thanks again for your review! We wanted to follow up to see if our responses helped to address your concerns.

---

> > > ### Comment · Reviewer_o3xo · 2022-08-09
> > > **To authors**
> > >
> > > Thanks for the detailed response. It answers my questions.

---

### Meta-Review · Area_Chair_hakr · 2022-08-26

**Recommendation:** Accept
**Confidence:** Certain

**Metareview:**

The paper studies online bipartite matching problem under a random graph model, and shows that using the expected mean of the degrees could achieve certain optimal performance under their graph model. The authors complement the theoretical study with empirical evaluation, and demonstrates that estimating the degrees would result in good performance in online bipartite matching.

The reviewers agree that the paper contains good technical contributions to the problem, and is worth to be published. There is some concern on whether the paper is related to ML/AI or is a purely combinatorial optimization paper. In particular, the reviewers share the concern that the theoretical algorithm takes the expected degrees as the input, and not the predicted degrees as suggesting an estimation with errors. After some discussions among the reviewers, here is my conclusion and recommendation:

1. The technical algorithm is like a combinatorial optimization algorithm, but it has a strong indication that degree estimation could be helpful in algorithm design. This is further complemented by the empirical study of the paper. Therefore, the study fits into the data-driven optimization and algorithm design paradigm that would interest the ML/AI community.

2. The use of the term "predicted degrees" in the title/abstract/intro is indeed misleading. It gives a strong impression that the algorithm is using a prediction (or estimation) that contains estimation error, but it actually does not use such predicted degrees, and instead using the expected degrees as input. Of course expected degrees are still not the actual random degrees but they are not in the normal sense the "prediction" result. I strongly suggest the authors to properly change the title/abstract/intro to more accurately reflect what they are doing and to reduce confusion (If 3 out of 4 reviewers raise this issue, plus I also has this concern, the authors should expect a significant confusion if they do not revise the presentation). The authors should clearly state that their theoretical algorithm is for expected degrees, and this may indicate that using predicted degrees may be helpful, but the latter is not part of the theoretical result. The following is my try on the title:

(Optimal) Online Bipartite Matching with Known Expected Degrees on Random Graphs --- A Theoretical Justification of the Effectiveness of Algorithms Based on Ordering of Predicted Degrees

It is certainly not elegant, but hopefully it suggests the authors to spend some effort to give a more accurate title/abstract/intro in their presentation.

Also, in terms of their theoretical result, their Appendix D does provide some result regarding the noise in the prediction. But it looks like the presented result is not in the normal sense of the prediction error between the prediction and the ground truth. I also suggest the authors to substantiate this part and perhaps move it into the main text so that the paper indeed has some treatment on predicted degrees.

Overall, with the above comments and suggestions, I believe the paper has good contributions to be accepted at NeurIPS.

**Award:**

No

---

### Decision · Program_Chairs · 2022-09-14

Accept